# Long-Term Wetland Monitoring Using the Landsat Archive: A Review

Quentin Demarquet *, Sébastien Rapinel , Simon Dufour and Laurence Hubert-Moy

Place du Recteur Henri Le Moal, LETG UMR 6554 CNRS, University of Rennes, 35000 Rennes, France
* Correspondence: quentin.demarquet@univ-rennes2.fr

**Abstract:** Wetlands, which provide multiple functions and ecosystem services, have decreased and been degraded worldwide for several decades due to human activities and climate change. Managers and scientists need tools to characterize and monitor wetland areas, structure, and functions in the long term and at regional and global scales and assess the effects of planning policies on their conservation status. The Landsat earth observation program has collected satellite images since 1972, which makes it the longest global earth observation record with respect to remote sensing. In this review, we describe how Landsat data have been used for long-term ($\geq$20 years) wetland monitoring. A total of 351 articles were analyzed based on 5 topics and 22 attributes that address long-term wetland monitoring and Landsat data analysis issues. Results showed that (1) the open access Landsat archive successfully highlights changes in wetland areas, structure, and functions worldwide; (2) recent progress in artificial intelligence (AI) and machine learning opens new prospects for analyzing the Landsat archive; (3) most unexplored wetlands can be investigated using the Landsat archive; (4) new cloud-computing tools enable dense Landsat times-series to be processed over large areas. We recommend that future studies focus on changes in wetland functions using AI methods along with cloud computing. This review did not include reports and articles that do not mention the use of Landsat imagery.

**Keywords:** ecosystem services; Landsat; time series; artificial intelligence; machine learning; cloud-computing; remote sensing; wetland functions

## 1. Introduction

Wetlands feature a wide variety of habitats, such as mangroves, swamps, fens, lagoons, and marshes, that provide many functions and ecosystem services [1]. However, over several decades, agriculture, urbanization, and climate change damaged approximately half of the wetland areas worldwide [2,3] even though many conservation policies have been implemented since the Ramsar Convention in 1974. Managers need tools to monitor wetlands and to assess the effects of planning policies on their conservation status [4,5]. However, wetlands are difficult to identify and characterize due to their dispersal [6], their fine-grained pattern [7], and the high spatio-temporal dynamics of the water and vegetation that shape them [8]. Thus, monitoring wetlands requires intra-annual observations of the Earth at high spatial resolutions for several decades and observations that cover large areas [9]. Moreover, while there have been global mapping initiatives such as the global wetland map developed by the Center for International Forestry Research (https://www2.cifor.org/global-wetlands/, accessed on 1 November 2022), the global map of saltmarshes [10], or the global map of peatlands [11], there is a gap with respect to long-term monitoring efforts.

The Landsat Earth observation program continuously acquired high-spatial-resolution images worldwide for 50 years [12]. Specifically, Landsat missions comprised five sensors launched since 1972. These sensors have collected images at different spatial resolutions in panchromatic (15 m), visible (30–80 m), near-infrared (30–80 m), mid-infrared (30–80 m),

and thermal infrared bands (60–120 m). The Landsat archive has been integrated into data collections, which improved the radiometric consistency and quality of images among Landsat sensors [13]. Images have been acquired with a wide swath (185 km) every 16 days by Landsat 1–7 and every 8 days by Landsat 8–9 [14]. More than five million images have been stored in the Landsat archive [15], including radiometrically and geometrically corrected products, cloud-free composites, and vegetation indices. The Landsat archive has been provided for free by the US Geological Survey (USGS) since 2008 [12].

Given the characteristics of the Landsat archive, the time series of Landsat images have been used widely for a variety of applications, including long-term wetland monitoring, but a global literature review on this topic remains lacking. In this context, this study aimed to provide comprehensive insight into the benefits and challenges of using the Landsat archive for long-term wetland monitoring. To this end, we performed a systematic review of the literature that focused on long-term wetland monitoring and Landsat data analysis issues. The term "wetland" used in this study is based on the Ramsar definition: areas of marsh, fen, peatland, or water; whether natural or artificial permanent or temporary; with water that is static or flowing, fresh, brackish, or salt, including areas of marine water the depth of which at low tide does not exceed six meters [16].

## 2. Methodology and Analysis

We applied the Preferred Reporting Items for Systematic Reviews and Meta-Analyses (PRISMA) method [17] (Figure 1). Specifically, the literature search of peer-reviewed articles in English was based on expert definitions of 49 keywords divided into three topics: "Landsat", "monitoring", and "wetlands" (Table S1). The period of the literature search ranged from 2009 (i.e., one year after the free release of Landsat images) to 1 March 2022. It was conducted globally to consider all wetland types. Only studies that monitored wetlands for at least 20 years were considered. Keywords in the title, abstract, or keyword fields were sought using queries of the Institute for Scientific Information Web of Science (WOS) and Elsevier Scopus databases. As a result, 401 and 459 articles were retrieved from WOS and Scopus, respectively. After removing duplicates, all titles and abstracts were read to remove irrelevant articles, resulting in a final selection of 351 articles (Figure 1, full list available in Table S2).

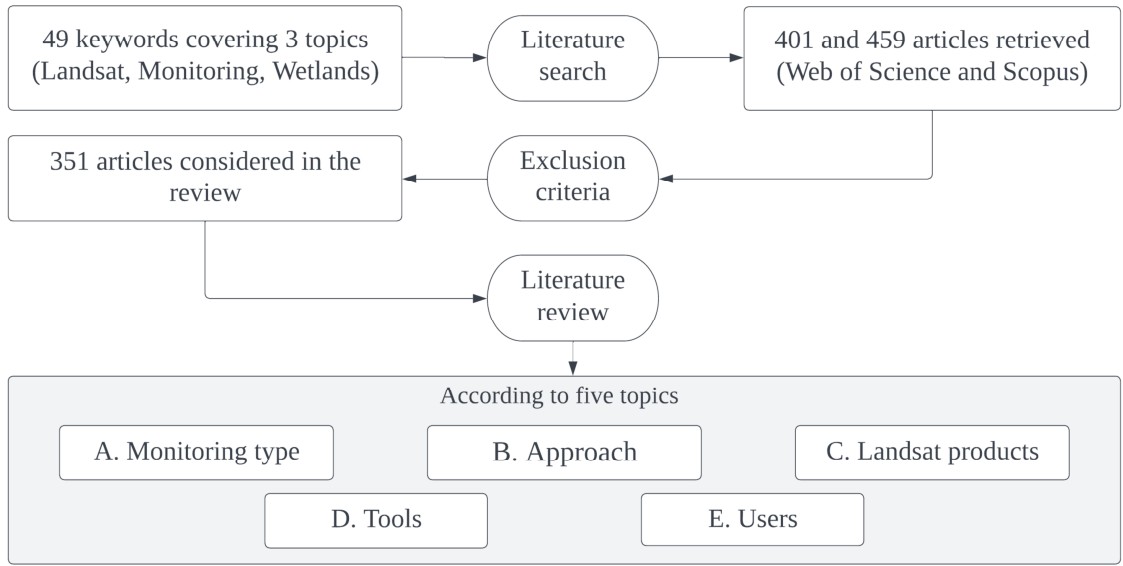

**Figure 1.** Flowchart used to search and review the literature on Landsat, monitoring, and wetlands (time span: 2009–2022).

The review was based on an analysis grid of 5 topics and 22 attributes qualified by categories (Table 1; see Supplementary Materials Table S3 for a full description of the categories). Attribute B1 "Wetland type" was characterized according to the Ramsar classifi-

cation system [16] (Supplementary Materials Table S4). When the wetland type was missing or uncertain in an article, Google Earth images were used to provide a visual interpretation of the studied wetland(s). When the wetland type changed over time, all wetland types in the article were considered. Attributes B6 and B7 were essential biodiversity variables (EBVs) and essential climate variables (ECVs), respectively. EBVs, which are spatial indicators recognized by the scientific community, characterize the ecosystem's structure and functions, species, and plant communities [18]. Similarly, ECVs address climate monitoring via atmospheric, land, and ocean observations [19]. From this perspective, EBVs and ECVs are valuable indicators for monitoring the functions and services provided by wetlands.

**Table 1.** Topics, attributes, and categories used in the literature review (see Supplementary Materials Table S3 for a full description of categories). LULC: land use/land cover change; FAPAR: fraction of absorbed photosynthetically active radiation.

| Topic | Attribute | Categories |
|---|---|---|
| A. Landsat data analysis | A1. Change detection | Diachronic; Multitemporal; Time series |
| | A2. Methods | Classification; Regression; Profile analysis |
| | A3. Artificial intelligence | Yes; No |
| | A4. Deep learning | Yes; No |
| | A5. Supervised method | Yes; No |
| | A6. Validation | Field data; Visual image interpretation; Both; None |
| B. Wetland monitoring | B1. Wetland type | Defined based on level 2 Ramsar classification |
| | B2. Spatial extent | Local; Regional; Continental; Global |
| | B3. Topic | LULC; Fragmentation and connectivity; Biophysical parameters; Hazards; Ecosystem services; Biodiversity; Hydrology |
| | B4. Drivers of change | LULC changes; Climate change; Invasive species; Restoration and/or conservation; Combination; No drivers of change |
| | B5. Essential Biodiversity Variables | Species population; Species traits; Community composition; Ecosystem function; Ecosystem structure |
| | B6. Remote Sensing—Essential Biodiversity Variables | Species phenology; Species morphology; Species physiology; Population structure by age/size class; Species distribution; Species abundance; Community diversity; Species composition; Ecosystem phenology; Ecosystem physiology; Ecosystem disturbances; Spatial configuration; Habitat structure |
| | B7. Remote Sensing—Essential Climate Variables | Lakes; Soil moisture; River discharge; Groundwater; Glaciers; Ice sheets and shelves; Snow cover; Permafrost; Albedo; Land use/land cover; Above-ground biomass; Land surface temperature; Evapotranspiration; Fire; Leaf area index; Soil carbon; FAPAR; Anthropogenic greenhouse gas fluxes; Human water use |
| | B8. Intra-annual observations | Yes; No |
| | B9. Study period | 20–30 years; 30–40 years; More than 40 years |
| C. Landsat products | C1. Satellite | Landsat 1-2-3 (MSS); Landsat 4-5 (MSS-TM); Landsat 7 (ETM+); Landsat 8 (OLI-TIRS) |
| | C2. Spectral domain | Visible; Infrared; Thermal; Combination |
| | C3. Pre-processing level | Level 1; Level 2; Derived products; Composite |
| D. Tools | D1. Cloud computing | Yes; No |
| | D2. Open-source software | Yes; No |
| E. Users | E1. Users | Scientists; Managers; Both |
| | E2. Journal discipline | Remote sensing; Geography; Earth and planetary science; Forestry; Aquatic, marine and water science; Environmental science; Sociology; Ecology; Multidisciplinary science; Computer science; Land management; Climatology |

Given the benefits of using cloud computing to analyze Landsat time series [20], we explored whether articles using this approach (attribute D1) provided new insights on long-term wetland monitoring. To this end, a *v*-test was calculated to compare the

distribution of each category (Table 1) in the articles based on cloud computing to the overall distribution in all articles [21]. An absolute *v*-test value of 1.96 corresponds to a *p*-value of 0.05. The higher the *v*-test value, the more the category was typical of using cloud computing. Statistical analysis was performed using R software [22] with the *FactoMine R* package [23]. It is important to note that this review focuses exclusively on Landsat imagery and did not include reports or articles that did not mention the use of Landsat imagery. Hence, some approaches developed for long-term wetland monitoring such as the evaluation of wetland habitat change that was conducted by the US Fish and Wildlife Service [24] were ignored.

## 3. Long-Term Wetland Monitoring

### 3.1. Spatial Extent and Global Distribution

The spatial extent of the reviewed studies varied by world region (Figure 2). Overall, most studies (73%) focused on the local scale, while only 27% focused on the regional scale. At the local scale, most studies were performed in Asia, especially China (15%) and India (8%), followed by Africa (8%), North America (5%), South America (4%), Oceania (3%), and Europe (2%). At the regional scale, a similar trend was observed, with most studies being performed in China (7%), followed by the United States (USA) (3%), India (2%), and Canada (2%). Three studies were performed at the continental scale, specifically North America [25], Central and South America [26], and Europe [27]. Conversely, no studies were performed on a global scale, although Schwatke et al. [28] monitored the extent of 32 lakes around the world. Comparing the locations of the studies to those of wetlands highlighted several gaps (Figure 2): Many large wetlands have not been monitored in the long term, particularly in South America, Equatorial Africa, Oceania, and Russia. Moreover, the comparison between the distribution of reviewed studies and wetland areas according to the world temperature domains (Figure 3) emphasizes that polar, cool temperate, and especially boreal domains were understudied. Conversely, warm temperate, sub-tropical, and tropical domains were covered by numerous long-term Landsat studies, although they were mostly focused on intertidal swamp forests (Figure 4).

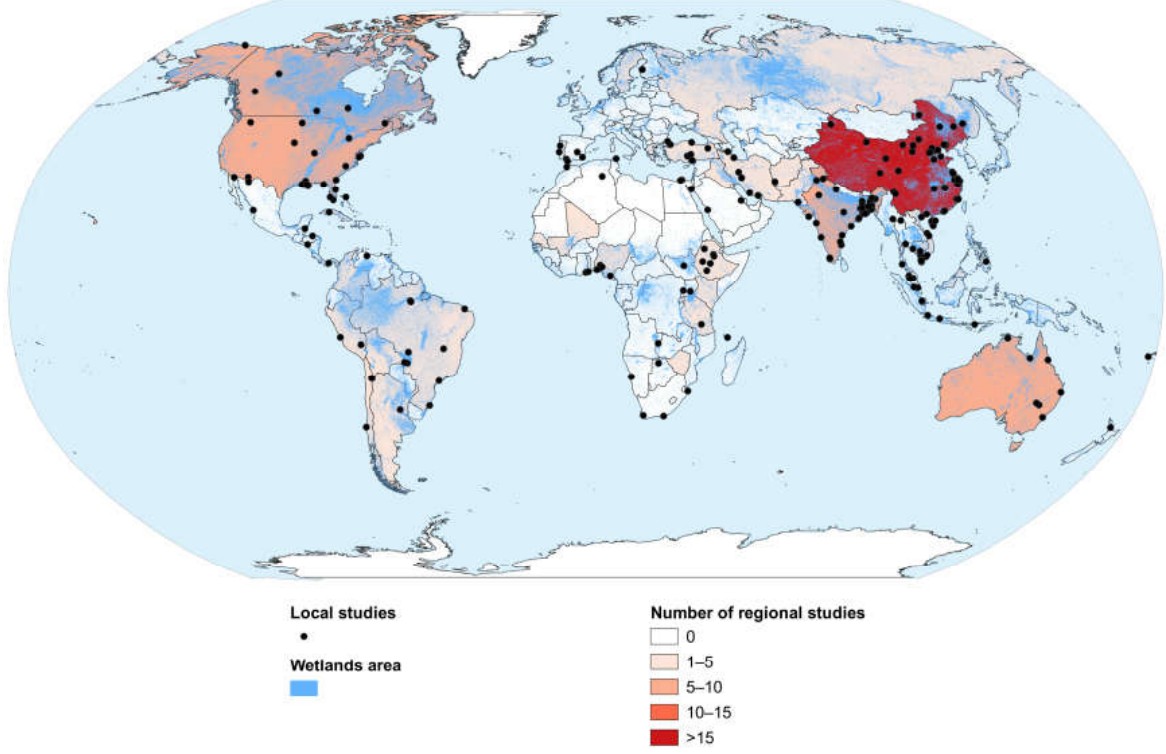

**Figure 2.** Spatial distribution of the reviewed studies. Source of the global wetland map: [29].

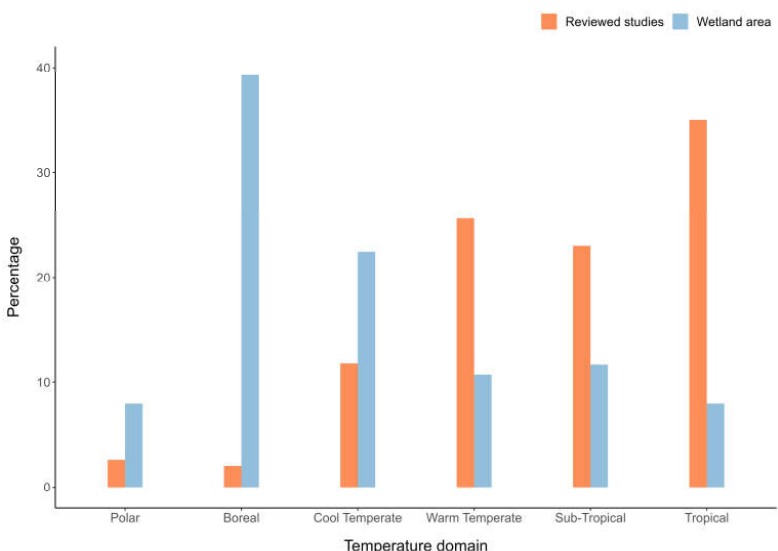

**Figure 3.** Distribution of the 351 reviewed studies and wetland areas [29] according to world temperature domains [30].

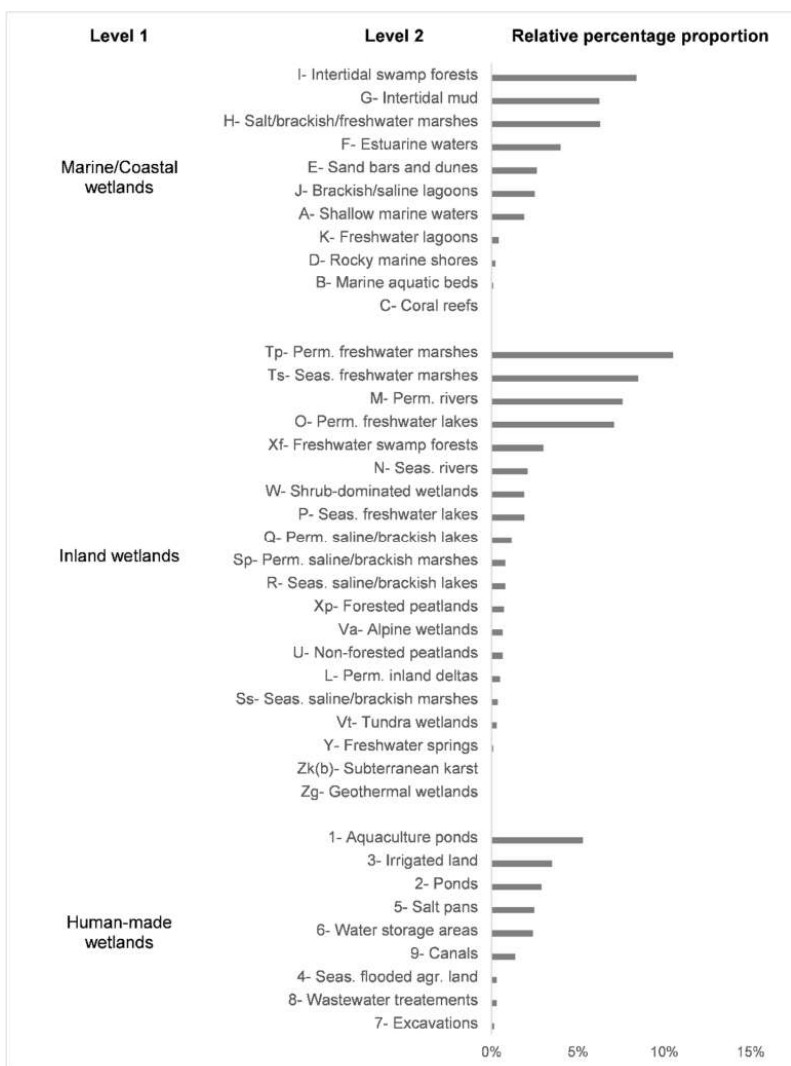

**Figure 4.** Percentage of wetland types investigated in the reviewed studies classified by levels 1 and 2 of the Ramsar wetland classification [16]. *Seas.*: seasonal; *Perm.*: permanent.

### 3.2. Wetland Types

The percentage of wetland types surveyed in this review varied by Ramsar classification (Figure 4). Globally (Level 1), inland wetlands (48%) and marine/coastal wetlands (33%) were studied the most, while human-made wetlands (19%) were studied the least. Among the marine/coastal wetlands, intertidal swamp forests (8%), intertidal mudflats (6%), tidal salt/brackish/freshwater marshes (6%), and estuarine waters (4%) were studied the most. Conversely, freshwater lagoons and marine aquatic beds were rarely studied, e.g., [31,32]. Among inland wetlands, permanent freshwater marshes (10.5%), seasonal freshwater marshes (8.5%), permanent freshwater rivers (8%), and lakes (7%) were studied the most. Conversely, many inland wetland types, such as tundra wetlands, inland deltas, and freshwater springs, were rarely studied (<1%) [33–35]. Among human-made wetlands, aquaculture ponds (5%), irrigated land (3.5%), ponds (3%), and salt pans (2.5%) were studied the most, while seasonally flooded agricultural land, which includes pastures, was rarely studied (e.g., [36]).

### 3.3. Long-Term Wetland Changes

The distribution of the reviewed studies according to categories of long-term wetland changes is shown in Figure 5.

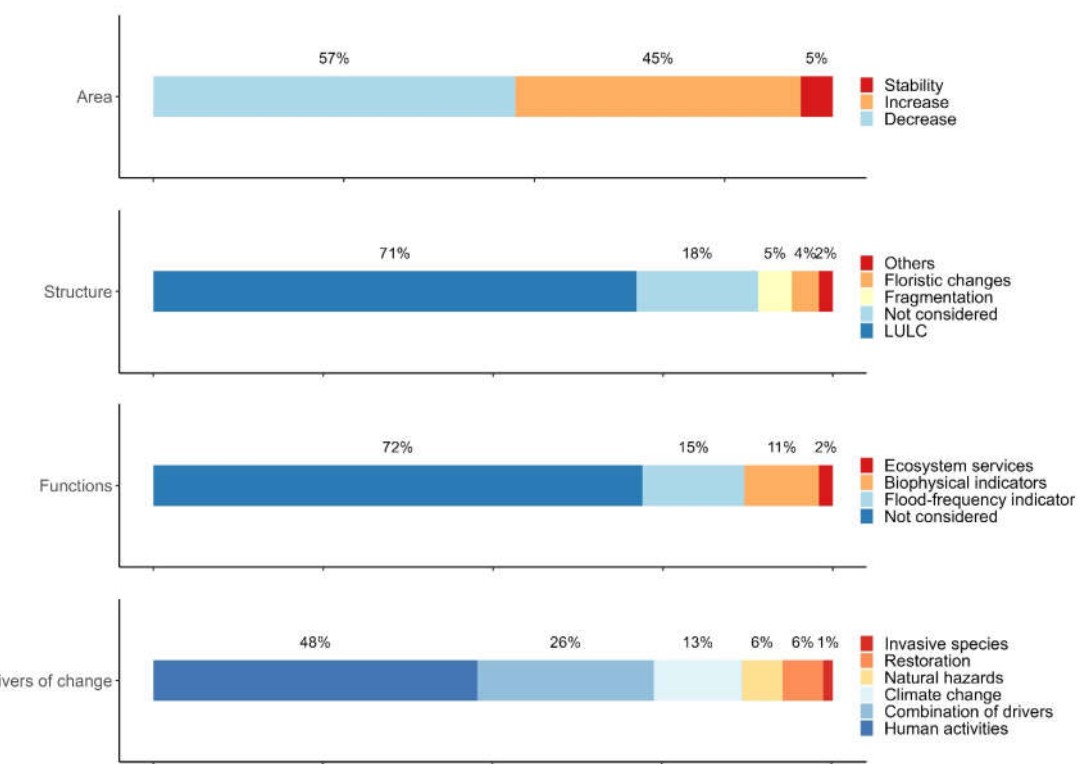

**Figure 5.** Distribution of the reviewed studies according to categories of long-term wetland change.

### 3.3.1. Area

A decrease in wetland area was the most frequent change reported worldwide (57% of the reviewed studies; Figure 5), including mangroves in West Africa [37], Belize [38], India [39], Vietnam [40], and Madagascar [41], as well as estuarine wetlands [42] and lakes [43] in India, wet forests in Nigeria [44], freshwater marshes in Uganda [45], and tundra wetlands in Canada [46]. Conversely, 45% of the studies reported an increase in wetland areas, including mangroves in China [47], Vietnam [48], and Australia [49], and alpine wetlands [50] and glacial lakes in the Andes Mountains [51]. To a lesser extent, wetland areas remained stable in 5% of the studies, including mangroves in Bangladesh [52] and sand bars in India [53]. Interestingly, Tiné et al. [54] described a long-term increase in

wetland areas in Quebec due to an increase in the beaver population, which created new hydrological conditions that promoted wetland expansion.

### 3.3.2. Structure

The fragmentation of wetlands was described in 5% of the reviewed studies (Figure 5). Most of these studies indicated an ongoing increase in wetland fragmentation over time, such as for waterbodies in India [55], the Prairie Pothole Region of North America [56], coastal wetlands in China [57], and Ghana [58], as well as lakes on the Tibetan Plateau [59]. Conversely, some studies emphasized an increase in wetland connectivity, such as for floodplains in Australia [60] and mangroves in Pakistan [61].

The long-term monitoring of floristic changes within wetlands was addressed in 4% of the studies. For example, O'Donnell and Schalles [62] showed a decrease in the biomass of *Spartina alterniflora (syn. Sporobolus alterniflorus)*, especially for small patches in salt marshes on the coast of Georgia, USA. Lucas et al. [63] indicated that variations in biomass were explained by the age of *Rhizophora* spp. in a mangrove in Malaysia. In the Sundarbans mangroves in Bangladesh, Ghosh et al. [64] described a decrease in *Heritiera fomes* and *Excoecaria agallocha* cover from 1977 to 2015 but an increase in *Ceriops decandra*, *Xylocarpus mekongensis*, and *Sonneratia apelatala* cover.

The other parameters of wetland structure were monitored less frequently. For example, an increase in above-ground mangrove biomass was measured in Vietnam [65], and the percentage of mangrove canopy cover was monitored in Australia [66]. The dynamics of natural habitats within wetlands were monitored in shallow lakes in Argentina [67], boreal wetlands in Canada [68], and floodplains in China [69].

### 3.3.3. Functions

Many of the reviewed studies derived functional indicators from the Landsat archive. A flood-frequency indicator was explored the most (15% of the studies; Figure 5). Overall, studies highlighted a decrease in flood frequency, such as in floodplain wetlands in India [70], in shallow intermittent lakes in Nebraska, USA [71], and in the Macquarie Marshes in Australia [72]. However, a few studies described an increase in flood frequency, such as in the northern part of the Doñana Marshes in Spain [73]. Interestingly, Cai et al. [74] described an alluvial wetland in Zambia for which floods had recently increased in the area, which disturbed wetland ecosystems and Indigenous populations. Biophysical indicators that characterize wetland functioning were derived from the Landsat archive in 11% of the studies. For example, the normalized difference vegetation index (NDVI) derived from Landsat time series revealed decreasing productivity in the Andes in southern Peru [75] but increasing productivity in the Tapacarí Province of Bolivia [76]. The biophysical properties of lakes have also been monitored worldwide: Pirali Zefrehei et al. [77] described increasing chlorophyll-a concentrations and turbidity in the Choghakhor wetland in Iran; Ho et al. [78] observed decreasing extents of the phytoplankton bloom from 1984 to 1991 and then a significantly increasing extent from 1991 to 2011 in Lake Erie (North America); Alademomi et al. [79] observed an increase in lagoon temperature in Nigeria using thermal Landsat bands. Less than 2% of the studies investigated changes in wetland functions and ecosystem services. For example, two studies used the NDVI trend to reveal a decrease in carbon storage in mangroves in Fiji [80] and in coastal wetlands in China [81]. Duncan et al. [82] used the NDVI trend in Cuba to explore the relation between mangrove productivity and extent, considering the habitat function for threatened mammal species. Interestingly, Ekumah et al. [58] described a decrease in coastal wetland ecosystem health in Ghana by combining functional indicators related to habitat fragmentation, vegetation phenology, flooding frequency, and land use/land cover (LULC) persistence. Changes in the monetary valuation of ecosystem services were investigated in a mangrove in Iran based on the NDVI trend [83] and in a lake in the Xinjiang region of China based on LULC changes [84].

### 3.4. Drivers of Change

3.4.1. Human Activities

The most common driver of wetland changes identified using the Landsat archive was LULC changes related to human activities (47.8%; Figure 5). Decreases in wetland areas were mainly due to urbanization (e.g., [85,86]), agriculture (e.g., [45,87]), and industrialization (e.g., [88]) or coastal land reclamation (e.g., [89]) to a lesser extent. The development of artificial wetlands (e.g., aquaculture ponds) is also responsible for the decrease in the area of natural wetlands worldwide, such as in China [90], Indonesia [91], Vietnam [92], India [93], and Egypt [94]. The consumption of natural resources related to water management (e.g., [95,96]) or deforestation [97] was also identified as a negative driver of change. Conversely, LULC changes may increase the area of natural wetlands, such as when a dam is constructed (e.g., [98,99]) or when farmland is abandoned [100].

3.4.2. Natural Hazards

Natural hazards were reported as a driver of change in wetlands in 6% of the reviewed studies. Several studies monitored post-fire conditions in wetlands: Darmawan et al. [101] observed the natural restoration of mangroves in Indonesia, and Mexicano et al. [102] described a significant increase in primary production the year after a fire. Other studies focused on the impacts of hurricanes on coastal wetlands: Carle et Sasser [103] observed a significant decrease in vegetation productivity in Wax Lake Delta (LA, USA), and Zhang et al. [104] observed that mangroves took 2–6 years to reach their full extent in the Biscayne National Park (FL, USA). Sedimentation is recognized as a key factor in increasing the area of wetlands in floodplains [33,105]. In addition, many studies highlighted erosion and accretion in coastal wetlands, such as in mangroves in Vietnam [106], as well as in lagoons in Namibia [107], China [108], and Turkey [109].

3.4.3. Climate Change

Climate change was identified as a driver of change in 13% of the reviewed studies. Changes in precipitation varied among climatic regions. In Australia, precipitation increased in the northeast tropical region, which increased the area of mangroves [49], but decreased in the southeast temperate region, which decreased the area of lakes [110]. The impact of rising temperatures in wetlands increased worldwide. For example, increased phenological activity has been observed in boreal regions of Canada in tundra [25] and peatlands [111] and in oases in the arid region of northwestern China [112]. In the tropics, rising temperatures have driven changes in the floristic composition of mangroves in Bangladesh [64]. Rising temperatures caused glaciers to melt, with new lakes appearing in the Andes cordilleras [50], and sea levels to rise, resulting in the loss of some coastal marshes in Virginia, USA [113].

3.4.4. Restoration and/or Conservation Policies

Overall, 6% of the reviewed studies used the Landsat archive to monitor the impacts of conservation or restoration policies. The studies showed positive effects, such as the restoration of lost wetlands [61,101], improved vegetation functioning [114,115], the maintenance of flood cycles [60,73], greater connectivity [60,61], and improved bird habitat function [116]. Most studies focused on mangroves, e.g., [117,118], but restoration or conservation policies also succeeded on floodplains [60,115], alpine wetlands [76], coastal marshes [73], and peat swamp forests [114].

3.4.5. Invasive Species

Invasions of alien plant species in wetlands were successfully observed using the Landsat archive. For example, the expansion of *S. alterniflora* was monitored in coastal wetlands [119,120], the effects of *Azolla filiculoides* growth on ecosystem services of lagoons were assessed [121], and the invasion of *Eichhornia crassipes* was mapped in lakes in Rwanda [122].

### 3.4.6. Combination of Drivers of Change

In the reviewed studies, 26% of wetland changes were due to a combination of several drivers (Figure 5). For example, water consumption by agriculture and industries, combined with a decrease in precipitation, resulted in a decrease in the area of wetlands in China [123]. In an arctic archipelago of Canada, climate change and the expansion of a bird species resulted in wetland degradation [46]. In addition, the combination of the expansion of invasive species and LULC changes decreased the area of water bodies [32,124]. Interestingly, many studies highlighted that a combination of drivers of change can both decrease and increase the area of wetlands. For example, LULC changes combined with sea level rise increased the area of salt marshes but decreased that of salt pans [125]. In China, Wu et al. [126] found that climate change combined with agricultural development could decrease or increase the area of wetlands depending on the conservation policy.

## 4. Landsat Data Analysis

The distribution of the reviewed studies according to categories of Landsat data analysis is shown in Figure 6.

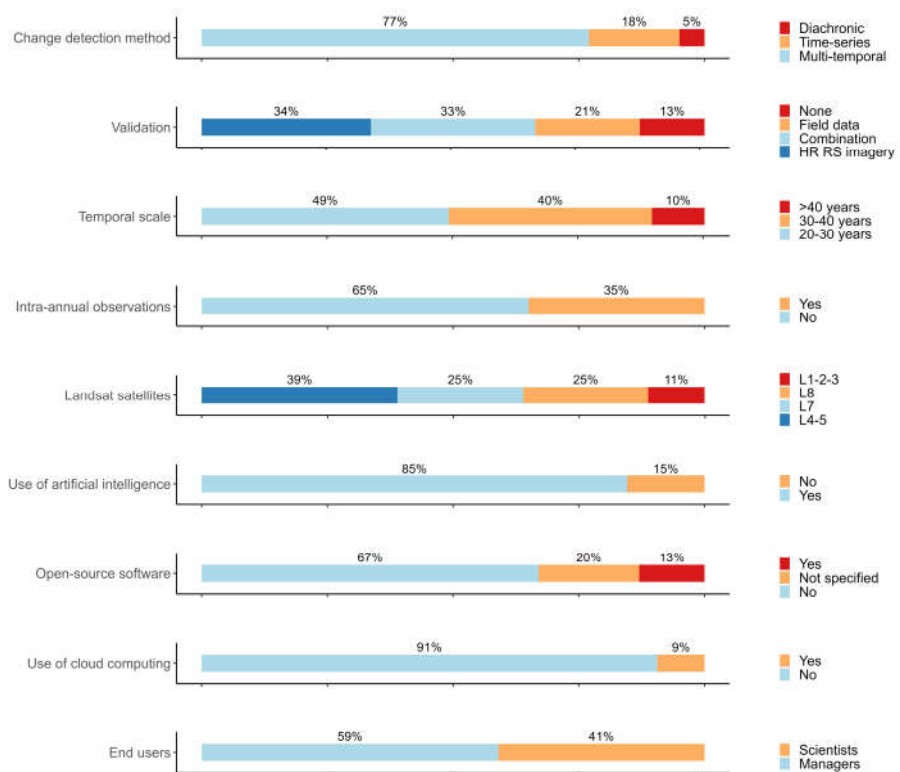

**Figure 6.** Distribution of the reviewed studies according to categories of Landsat data analysis.

### 4.1. Change Detection Method

Changes in wetlands were detected mainly using multi-temporal approaches (77.5%), followed by time-series analyses (18%), e.g., [127,128], and the diachronic approach (5%), e.g., [129,130] (Figure 6). The diachronic approach (change detection by processing two images of different dates) widely used to map LULC, although fast and simple, does not consider the intra- and inter-annual variability of wetland water and vegetation, which leads to errors in wetland area estimation and habitat classification. While the multi-temporal approach (processing of more than two images for several different dates), which is also fast and simple to implement, improves the results obtained with the diachronic approach, only remote sensing data time series (processing of most available images) that consider the intra- and inter-annual variability of wetland water and vegetation can help

assess wetland functions. However, the use of time series poses the challenge of developing methods that are able to process such amounts of data.

### 4.2. Validation

Landsat-derived maps and products were validated mainly by visual interpretation of high-resolution remote sensing imagery (34%), such as those available on Google Earth. In 33% of the studies, visual interpretation was combined with field data. Only 21% of the studies used field data for validation [41,131]. Surprisingly, ca. 13% of the studies did not mention a validation procedure (Figure 6).

### 4.3. Temporal Scale of Studies

The study period was 20–30, 30–40, and more than 40 years for 49.3%, 40.5%, and ca. 10.5% of the reviewed studies (Figure 7). Notably, Kawser et al. [132] analyzed the impacts of the great Assam earthquake (1950) on an estuarine floodplain in Bangladesh by processing historical toposheets combined with the Landsat archive from 1942 to 2019. Gopalakrishnan et al. [133] explored changes in mangroves in Peninsular Malaysia using historical records and the Landsat archive for a 74-year period (1944–2018).

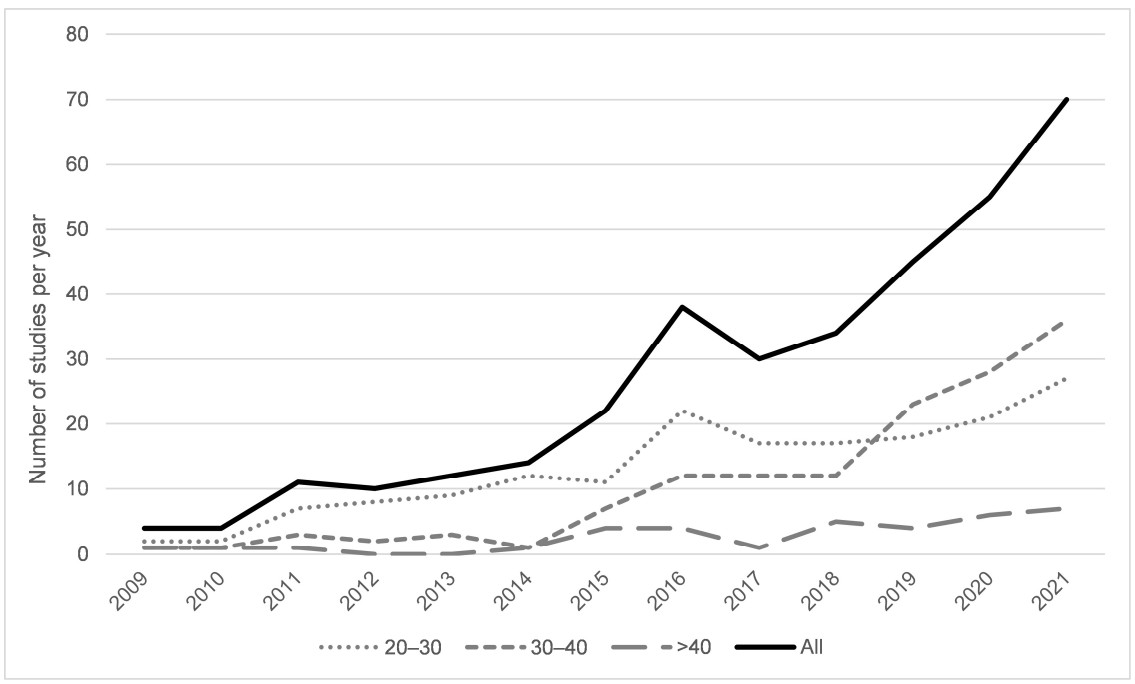

**Figure 7.** Number of reviewed studies published per year from 2009 to 2021 by the duration of the study period (years).

From 2009 to 2021, the number of studies published per year with periods of 20–30, 30–40, and more than 40 years increased from 2 to 27, from 1 to 36, and from 1 to 7, respectively (Figure 7).

### 4.4. Intra-Annual Observations

Only 35% of the reviewed studies used intra-annual observations (Figure 6). Notably, Borro et al. [134] mapped shallow lakes in South America from 1987 to 2010 using 1–8 images from Landsat 5 and 7 per year. Ouyang et al. [135] investigated the impacts of drought on the extent of surface water in the USA from 1984 to 2011 using 1–5 images from Landsat 5 and 7 per year. Other studies used more images per year, such as Liang et Li [136], who investigated changes in lakes in China from 1988 to 2014 using 12 images from Landsat 5, 7, and 8 per year [136].

### 4.5. Use of Landsat Sensors

All Landsat sensors were used in the reviewed studies but in different percentages (Figure 6). Landsat 4–5 (MSS/TM) images were used the most (38.9% of the studies), while Landsat 1-2-3 (MSS) images were used the least (11.2%). Landsat 7 ETM+ and Landsat 8 OLI/TIRS images were used in 25% and 24.8% of the studies, respectively. Visible and infrared bands were used in 49% of the studies. Conversely, only 2% of the studies used thermal bands, e.g., [137,138].

The Landsat bands were used at different pre-processing levels. While 20.5% of the studies did not specify a pre-processing level, most (63%) used Level 1 (radiometrically and geometrically corrected) Landsat data. Conversely, 16% of the studies used Level 2 (atmospherically corrected) Landsat data. Fewer than 1% of the studies used Landsat composites [139,140].

The combination of the Landsat archive and data derived from other remote sensing sensors was shown in 34.7% of the reviewed studies. Various topographic information derived from DEM data can be combined with spectral data to capture additional wetland characteristics not captured by spectral data alone [141]. More precisely, 17.4% of the reviewed studies used DEMs derived from remote sensing data, such as SRTM, e.g., [122]; ASTER, e.g., [51]; TanDEM-X, e.g., [63]; ALOS PALSAR, e.g., [63]; or LiDAR, e.g., [142], to characterize topography, vegetation, or water height. In 19.7% of the reviewed studies, optical or radar variables from other sensors such as MODIS, e.g., [119]; AVHRR [25]; ALOS AVNIR, e.g., [143]; IRS-LISS II/III/IV, e.g., [42]; SPOT 2/4/5, e.g., [144]; or Sentinel-1/2, e.g., [145], were combined with Landsat data. In 2% of the cases, studies used both DEMs and spectral variables, e.g., [146].

### 4.6. Use of Artificial Intelligence

Artificial intelligence (AI) was used in 84.6% of the reviewed studies for the classification (64.3%), regression (15.9%), and profile analysis (4.4%) of Landsat data (Figure 6). Supervised classification was the approach used most frequently. For example, Parihar et al. [147] classified Landsat MSS and TM data from 1973 to 2010 in West Bengal, India, applying a supervised maximum likelihood classifier, one of the classification methods used the most in the studies. Their classifications had an overall accuracy of 73.8–79.3%. Han et al. [148] investigated changes in vegetation in Poyang Lake, China, from 1973 to 2013 using inter-annual classification applied to data from Landsat 1-2-3 to Landsat 8. A comparison of several supervised classification algorithms showed that support vector machine (SVM) had the highest accuracy, with a Kappa index of >0.9. More recently, Berhane et al. [149] used the random forest algorithm to monitor LULC in the wetlands of the Midwestern USA. They classified 1749 Landsat data with an overall classification accuracy of 81–89%. The studies also used unsupervised classification methods. For example, Dewidar [150] classified Landsat MSS, TM, and ETM+ data by using the ISODATA algorithm to monitor temporal changes in the area of water in lagoons in Egypt from 1972 to 2007, with an overall accuracy of ca. 97.5%. Fitoka et al. [100] mapped wetland areas in Greece by combining Sentinel-2 and Landsat 5 TM data from 1986 to 2017. They combined classification trees with expert rules to distinguish 31 wetland classes with an overall accuracy of more than 90% over a total area of more than 2 million ha.

Raynolds and Walker [151] applied regression methods to Alaska, USA, from 1985 to 2011. They calculated and estimated the NDVI trend across the study area and observed a significantly negative regional trend in NDVI, mainly due to the development of the oil industry. Dingle Robertson et al. [152] applied spectral mixture analyses to 26 Landsat 5 TM images from 1984 to 2010 in Canada. The results showed a decrease in wetland areas, mainly due to the extension of recreational and agricultural activities and encroaching development. Regression models can also provide information on abiotic and anthropogenic factors that influence wetlands. For example, the Breaks for Additive Season and Trend algorithm was used for long-term monitoring [153] to identify global temporal trends in

wetlands (e.g., vegetation greenness) and provide information on seasonal components and abrupt changes.

Lv et al. [154] investigated wetland loss in a region of China from the 1980s to 2015 by performing a profile analysis of 3 spectral indices (the normalized difference vegetation index (NDVI), the normalized difference water index (NDWI), and the soil moisture monitoring index (SMMI)) derived from 27 Landsat images. Results showed a decrease in the wetland area of ca. 13.8% over three distinct periods. Among studies that used machine learning, nine used deep-learning algorithms. For example, Lin et al. [155] investigated changes in wetlands in Shanghai, China, from 1986 to 2013 using the Kernel Extreme Learning Machine (K-ELM) algorithm on six Landsat 5 TM images and compared its accuracy to those of Extreme Learning Machine, MLC, and SVM algorithms. Results showed that K-ELM performed the best, with an overall accuracy of 86%. Ehsani and Shakeryari [156] used a fuzzy ARTMAP neural network to process Landsat MSS, TM, ETM+, and OLI images to investigate changes in wetlands from 1977 to 2014. They observed a decreasing trend in wetland area, with a Kappa index of 0.84–0.89.

AI was also used to detect and remove pixels from cloudy Landsat images and then fill these gaps with other images. It achieved denser cloud-free time series that improved modeling quality [20]. For example, Schwatke et al. [28] increased the accuracy of spatial models of lakes by 41% by developing an automated tool to extract water areas and that detects and removes clouds in Landsat images and fills in some gaps with Sentinel-2 images.

Conversely, 15.7% of the studies did not use AI approaches, such as visual interpretation and digitization or band thresholding. For example, Xie et al. [157] investigated LULC changes in a nature reserve in China from 1988 to 2008 based on the visual interpretation of Landsat TM data. Chen et al. [158] applied Otsu's method to obtain an optimal threshold of Landsat bands to classify and monitor mangrove changes in Honduras from 1985–2013 (Kappa index = 0.82). Almahasheer et al. [159] thresholded NDVI to monitor mangroves on the coast of the Red Sea from 1972 to 2013, with 91% overall accuracy.

### 4.7. Open-Source Software

The Landsat archive has been processed less often using open-source software (13% of the reviewed studies; Figure 6). For example, Otero et al. [160] created maps of changes in mangrove cover in Malaysia from 1988 to 2015 from Landsat TM, ETM+, and OLI using R software. Tin et al. [161] processed and classified Landsat images using QGIS software to describe historical changes in mangroves in Vietnam from 1973 to 2016. Notably, Schaffer-Smith et al. [162] investigated dynamics of the area of water in wetlands in California, USA, from 1983 to 2015. They analyzed Landsat 4, 5, 7, and 8 data using Python to pre-process images and R software to classify images and assess accuracy, along with commercial ArcGIS software. Notably, 20% of the studies did not name which software they used.

### 4.8. Cloud Computing

Overall, 9.4% of the reviewed studies used cloud computing for long-term wetland analysis, of which all used the Google Earth Engine (GEE) cloud-computing platform [163] (Figure 6). Comparing cloud-computing studies to local-computing studies revealed statistically significant differences in several attributes (Table 2). Studies that used cloud computing performed more change detections using time series and thus used more intra-annual images. Similarly, they combined Landsat TM, ETM+, and OLI/TIRS data more often and at pre-processing Level 2. Cloud-computing studies were conducted more often at regional and continental scales than at the local scale. Regarding study periods, 30–40-year periods were more common for cloud-computing than local-computing studies. Cloud-computing studies did not use more open-source software than local-computing studies did. Cloud-computing studies were published more often (+17.8%) in remote sensing journals.

**Table 2.** Frequency (percentages) of attributes in studies that used Google Earth Engine that differed significantly ($|v\text{-test}| > 1.96$) from the frequency in all studies reviewed. "Class": Studies that used Google Earth Engine; "Class/Mod": studies with the attribute and belonging to the class; "Mod/Class": studies belonging to the class and with the attribute; "Overall": frequency of the attribute in all studies, regardless of the class; EBVs: essential biodiversity variables.

| Attribute | Class/Mod | Mod/Class | Overall | $v$-Test |
|---|---|---|---|---|
| Change detection using time series | 38 | 73 | 18 | 7.3 |
| Use of intra-annual observations | 23 | 85 | 34 | 6.2 |
| Level 2 of pre-processing | 31 | 51 | 15 | 5.1 |
| Use of L4-5 TM AND L7 ETM+ AND L8 OLI/TIRS images | 18 | 51 | 27 | 3.1 |
| Regional-scale study | 18 | 48 | 25 | 3.0 |
| Open-source software not used | 12 | 85 | 65 | 2.6 |
| Continental-scale study | 50 | 9 | 2 | 2.5 |
| Remote sensing journal | 17 | 40 | 22 | 2.4 |
| Study period of 30–40 years | 14 | 61 | 40 | 2.4 |
| Ecosystem function EBVs | 24 | 15 | 6 | 2.0 |
| Forces driving change not specified | 21 | 18 | 8 | 1.99 |
| Spectral domain of the Landsat archive not specified | 0 | 0 | 9 | −2.0 |
| Local-scale study | 5 | 39 | 73 | −4.2 |
| Change detection using multi-temporal images | 3 | 27 | 77 | −6.4 |

Several studies used GEE. For example, Cavanaugh et al. [26] investigated the sensitivity of mangrove extent relative to climate variability from the Atlantic coast of North America to the Pacific coast of South America from 1984 to 2011 using the Enhanced Vegetation Index from Landsat 5 TM and Landsat 7 ETM+ images available in GEE. Laengner et al. [27] used GEE to investigate changes in the area of salt marshes at the continental scale at 125 sites of ca. 20 km$^2$ each across Europe from 1986 to 2010. Shi et al. [146] assessed the impacts of historical changes on wetland areas in Zhenlai County (5350 km$^2$), China, from 1985 to 2018 using 2562 images and spectral indices including NDVI, enhanced vegetation index (EVI), land surface water index (LSWI), normalized difference water index (NDWI), and modified normalized difference water index (mNDWI) derived from Landsat 5 TM, Landsat 7 ETM+, and Landsat 8 OLI/TIRS surface reflectance and combined with Sentinel-1 SAR and Sentinel-2 MSI data. Wang et al. [145] described spatio-temporal variations in coastal wetlands in China from 1984 to 2018 using 62,000 images from Landsat 5 TM, Landsat 7 ETM+, and Landsat 8 OLI/TIRS.

*4.9. End Users*

The results of the reviewed studies targeted mainly the scientific community and stakeholders/policy makers (59%), providing information for management and economic and social policies that protect wetlands and/or improve wetland conservation (Figure 6). The remaining studies (41%) targeted only scientific communities primarily for modeling and for developing basic knowledge to better understand spatio-temporal dynamics of wetlands [164,165].

**5. Progress and Recommendations**

*5.1. Using the Landsat Archive for the Extensive Monitoring of Wetland Extent and Type*

This review highlights that wetlands have been monitored worldwide over large areas, ranging from local to continental scales. However, certain areas with high wetland density, such as boreal wetlands in Russia, tropical inland wetlands in Africa, and cool temperate wetlands in Europe, were monitored less often in the long-term and should be studied more extensively.

Nearly all wetland types in the Ramsar level 2 classification system were monitored using the Landsat archive. Mangroves and inland freshwater marshes were the types monitored the most, but rarer wetland types such as lagoon seabed vegetation [31] and freshwater springs [33] were also monitored. However, other wetland types that are widely

distributed around the world have been monitored less often and should be studied more in the future. This includes tundra wetlands, which cover ca. 25% of the global area of lakes [166] and face major conservation issues related to climate change [167].

*5.2. Using the Landsat Archive to Improve the Monitoring of Wetland Areas, Structure, and Functions*

5.2.1. Landsat Archive Enables the Monitoring of Wetland Area, Structure, and Functions

Changes in wetland areas, structure, and functions must be monitored to assess the conservation status of these ecosystems [168]. This review highlights that the Landsat archive is suitable for this purpose: (1) Many studies highlighted changes in wetland area; (2) changes in wetland structure were described using multiple indicators related to fragmentation, floristic composition, biomass, and percentage canopy cover; (3) changes in functions were described using indicators related to flooding frequency, vegetation productivity, chlorophyll-a concentration, and surface temperature. Some studies emphasized changes in functions such as carbon storage or support for bird habitats. This review revealed that the Landsat archive can also be used to identify drivers of wetland changes over long-term periods of time. Unsurprisingly, human activities and climate change are the main threats to wetland conservation. Interestingly, the long-term effectiveness of conservation and restoration policies has been successfully measured using the Landsat archive.

5.2.2. The Need for Crosswalks to Common Operational Frameworks

This review highlights advantages for the scientific community of using the Landsat archive for long-term wetland monitoring; however, wetland structure and functions are rarely considered in assessments of wetland conservation status [169]. This may be due to several reasons, including a lack of correspondence between indicators derived from the Landsat archive and those developed from common operational frameworks based on in situ observations. For example, indicators of area, structure, or functions identified from the Landsat archive were frequently similar to most EBVs (Table 3) and ECVs (Table 4). Correspondence with EBVs and ECVs should be explicitly mentioned in future research to provide better information on the potential to derive and monitor these indicators using the Landsat archive.

**Table 3.** Number and percentage of reviewed studies by remote sensing essential biodiversity variable (RS-EBV) as defined by [18]. The total number of studies exceeds 351 because some studies mentioned several EBVs.

| Class | RS-EBV | Number of Studies | Percentage of All Reviewed Studies | Example References |
|---|---|---|---|---|
| Species traits | Species phenology | 0 | 0% | |
| | Species morphology | 0 | 0% | |
| | Species physiology | 1 | 0.2% | [62] |
| Species population | Population structure by age/size class | 2 | 0.4% | [62,63] |
| | Species distribution | 0 | 0% | |
| | Species abundance | 0 | 0% | |
| Community composition | Community diversity | 0 | 0% | |
| | Species composition | 8 | 1.9% | [64,120–122,142, 155,170,171] |
| Ecosystem functioning | Ecosystem phenology | 7 | 1.8% | [58,119,172–174] |
| | Ecosystem physiology | 33 | 7.9% | [73,104,111,115] |
| | Ecosystem disturbances | 8 | 2.0% | [101,102,175,176] |
| Ecosystem structure | Spatial configuration | 25 | 6.1% | [59,177–179] |
| | Habitat structure | 326 | 79.7% | [39,180–183] |



**Table 4.** Number and percentage of reviewed studies by remote sensing essential climate variable (RS-ECV) as defined by GCOS (2010) [184]. The total number of studies exceeds 351 because some studies mentioned several ECVs. FAPAR: Fraction of absorbed photosynthetically active radiation.

| Domain | RS-ECV | Number of Studies | Percentage of All Reviewed Studies | Example References |
|---|---|---|---|---|
| Hydrosphere | Lakes | 57 | 14.6% | [136,185] |
| | Soil moisture | 20 | 5.1% | [34,72] |
| | River discharge | 0 | 0% | |
| | Groundwater | 0 | 0% | |
| Cryosphere | Glaciers | 2 | 0.4% | [50,51] |
| | Ice sheets and shelves | 0 | 0% | |
| | Snow cover | 0 | 0% | |
| | Permafrost | 0 | 0% | |
| | Albedo | 0 | 0% | |
| Biosphere | Land use/land cover | 278 | 71.1% | [107,125,186] |
| | Above-ground biomass | 3 | 0.8% | [63,65] |
| | Land surface temperature | 2 | 0.5% | [79,187] |
| | Evapotranspiration | 2 | 0.5% | [102,188] |
| | Fire | 0 | 0% | |
| | Leaf area index | 0 | 0% | |
| | Soil carbon | 0 | 0% | |
| | FAPAR | 0 | 0% | |
| Anthroposphere | Anthropogenic greenhouse gas fluxes | 0 | 0% | |
| | Human water use | 0 | 0% | |
| No ECVS in the reviewed study | | 27 | 6.9% | |

While this review shows that functional indicators have been derived from the Landsat archive, it also highlights that wetland functions and ecosystem services have rarely been assessed in long-term monitoring, and when they have, the assessment used an expert-based approach. Future studies should assess changes in wetland ecosystem functions and services using recognized and operational approaches, such as functional assessment [189] or the Rapid Assessment of Wetland Ecosystem Services [190]. However, it should be kept in mind that the spatial resolution of Landsat is insufficient for delineating small wetlands and/or small hydrogeomorphic units within wetlands and for assessing their functions. This is especially true for Landsat MSS images that have a spatial resolution of 80 m. In these cases, it is necessary to use a field approach. In addition, some wetland functions cannot yet be assessed from space because canopies prevent the direct observation of important features such as soil bacteria.

*5.3. Extend the Monitoring Period Backwards and Forwards in Open Access*

This review highlights that the Landsat archive can be used to map long-term and fine-grained changes in patterns, such as the spread of invasive species or the phenological response of vegetation after natural hazards, which would have been difficult to observe over shorter periods and/or at a larger scale. Although the Landsat archive has been available for 50 years, most studies that used it, including the most recent ones, covered 20–30 years. This could be due to the challenge of combining Landsat MSS images, which have different spatial and spectral resolutions than Landsat TM, ETM+, and OLI images. However, MSS images provide the ability to look back an additional decade, which would be valuable for improving the understanding of responses of wetlands to human development and climate change. Future research should extend the temporal monitoring period to at least 50 years by including not only new Landsat 9 images but also older MSS images, although some wetlands may not be detected and monitored due to the broad spatial resolution of these data.

The combination of topographic data with spectral data significantly improves wetland delineation, and several high spatial resolution global DEMs are available in open access [191]. Moreover, the future monitoring of wetland structure and function could be improved by using new variables such as the global canopy height model derived from GEDI and Landsat data [192] or Sentinel-1 interferometric coherence time series [193].

The large number of long-term wetland monitoring studies in this review from 2009 to 2022, especially in developing countries, shows the positive effect of providing open access Landsat archive data for wetland conservation worldwide, especially in developing countries where managers have limited resources and funds for monitoring highly threatened wetlands [4]. In this context, we remind practitioners that they should always acknowledge the USGS for making the Landsat data freely available.

### 5.4. The Era of AI and Cloud Computing

5.4.1. Progress in AI and Cloud Computing Provide New Opportunities for Long-Term Wetland Monitoring

This review illustrates how recent advances in AI (e.g., cloud masking, classification, and regression) have been used to analyze Landsat archive data automatically. Nevertheless, two points of progress must be discussed. The first concerns the use of reference data, which remain challenging to collect in some regions and/or for the past. This may explain why the accuracy of indicators derived from earlier Landsat images was rarely assessed. Future studies could make greater use of archive field databases, such as the Global Biodiversity Information Facility [194] for flora and fauna or the GlobalSoilMap [195] for soils, to further calibrate and validate models. As an alternative to field surveys, several tools such as TimeSync and CollectEarth support the collection of reference points via the visual interpretation of high-spatial-resolution images [196].

The second point concerns the analysis approach used. This review found that most studies used only cloud-free Landsat images, which discards a large amount of imagery, and that each year was modeled individually, which increases uncertainty in detecting change. Future research should focus on modeling the temporal profile using most images available in the Landsat archive. Cloud removal methods are now available that allow denser Landsat time series to be used over the entire globe, which can improve long-term monitoring, including in tropical areas. For example, Arévalo et al. [197] recently developed a set of tools in Google Earth Engine for this purpose. We calculated the normalized difference vegetation index (NDVI) time series in the Iraqi Hammar Marsh (Ramsar site ID 2242) from the Landsat archive for the period 1984–2022 as an example (Figure 8). Filtering Landsat data contributes to maintaining a high-quality time series, because it reduces the redundancy of information, residual noises in the images, and processing time and space.

This review also shows the relevance of cloud computing for processing the Landsat archive over large areas [14]. Cloud computing is an opportunity to reduce the technology gaps between countries in the Global North and South. All studies that used cloud computing chose the GEE for its easy access and use and for the availability of the entire Landsat archive pre-processed for surface reflectance (level 2). However, future studies could also use the lesser-known NASA Earth Exchange [198] or Microsoft Planetary Computer platforms, which are open access alternatives to GEE.

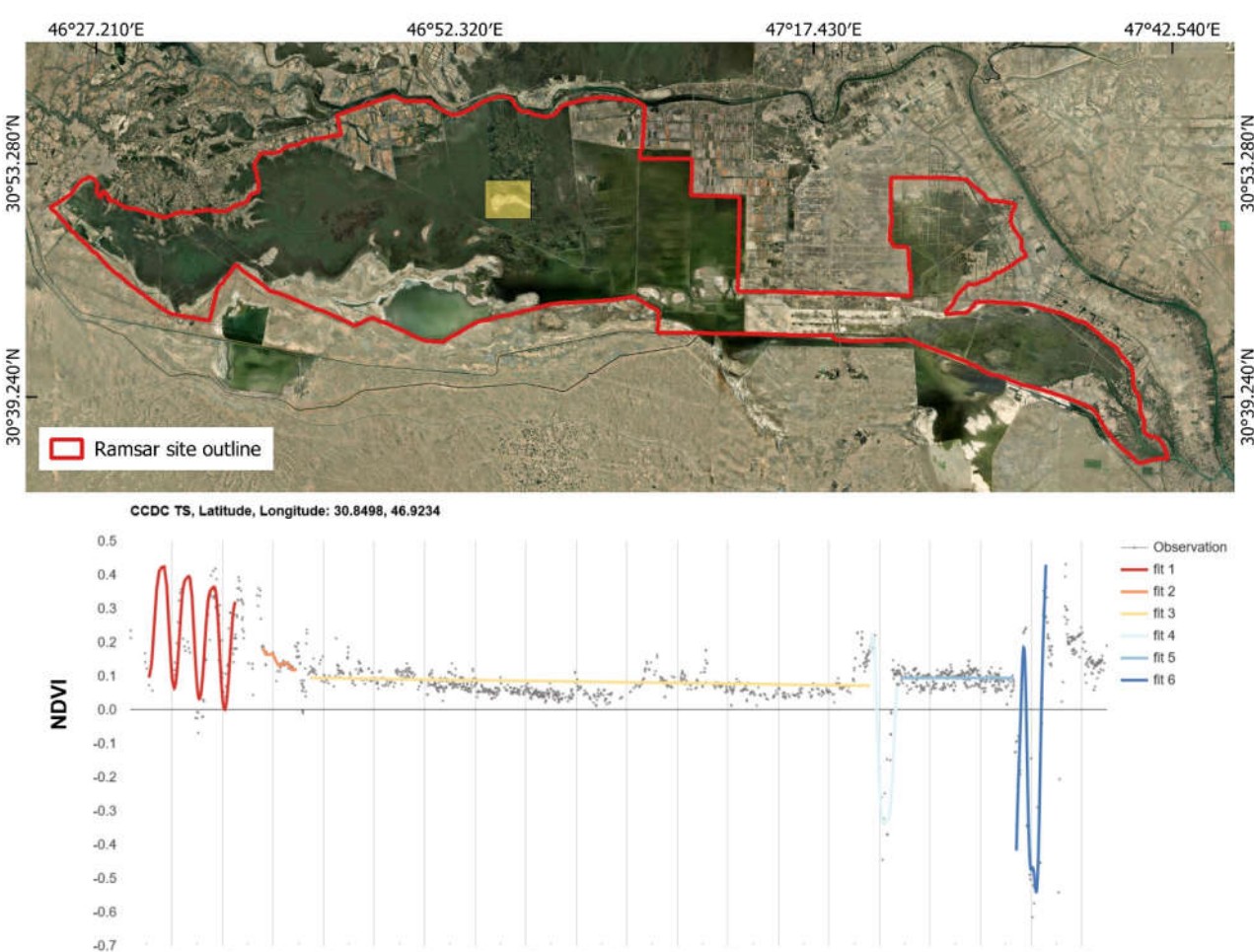

**Figure 8.** Example of the normalized difference vegetation index (NDVI) time series for a pixel (yellow rectangle) in the Iraqi Hammar Marsh (Ramsar site ID 2242) derived from the Landsat archive from 1984 to 2022 obtained using the CCDC algorithm developed by [197] in Google Earth Engine. Grey dots indicate Landsat observations available in the pixel and used for harmonic regression of the time series. The fits in the colored lines indicate time segments detected by the CCDC algorithm: a change in color indicates a change in land cover/land use.

### 5.4.2. Bridging the Gap between Remote Sensing and Wetland Monitoring

This review emphasizes that the analysis and interpretation of the Landsat archive differs among scientific disciplines. Studies published in environmental journals used field observations to provide new insights on wetland ecosystem functioning but rarely used the most advanced AI or cloud-computing tools. Conversely, studies published in remote sensing journals used innovative methods to analyze large datasets from the Landsat archive, but the changes detected were rarely interpreted or validated using the visual interpretation of high-spatial-resolution images. Thus, as described by Pettorelli et al. [199], scientists in environmental and remote sensing communities need to share skills and knowledge even more to facilitate and improve ecosystem monitoring using methods known to be effective. In this context, using open-source solutions provides more effective ways to share methods and thus increases reproducibility by providing widely available methods.

## 6. Conclusions

This review highlighted that the Landsat archive was used in hundreds of scientific studies for the global long-term monitoring of wetlands, especially in developing countries where managers have limited resources and funds for monitoring highly threatened wet-

lands. The continuity of Landsat missions since 1972 enables the monitoring of wetland areas, structure, and functions including fine-grained changes in patterns, which would have been difficult to observe over shorter periods and/or at a broader spatial resolution. This review also emphasizes that new cloud-computing tools enable dense Landsat times series to be processed over large areas. Future research should study more extensively certain areas or wetland types that are still poorly explored, extend the monitoring period backwards and forwards, make greater use of archive field databases to further calibrate and validate models, focus on modeling the temporal profile using most of the images available in the Landsat archive, crosswalk to common operational frameworks such as functional assessment procedure or essential biodiversity variables, and bridge the gap between environmental and remote sensing scientific communities.

**Supplementary Materials:** The following supporting information can be downloaded at https://www.mdpi.com/article/10.3390/rs15030820/s1. Table S1: Keywords selected per topic for the literature search. An asterisk (*) was used to retrieve variations of a term. Table S2: full list of reviewed articles. Although several references cited below are present in the manuscript, the following numbers do not match those provided in the manuscript. Table S3: Full description of the categories listed in Table 1. Table S4: Ramsar Classification System of wetlands. A single asterisk (*) indicates floodplain wetlands, such as seasonally flooded grasslands (including natural wet meadows), shrublands, woodlands, or forests. Double asterisks (**) indicate intensively managed or grazed wet meadows or pastures.

**Author Contributions:** Conceptualization, S.R., L.H.-M. and Q.D.; methodology, S.R., L.H.-M. and Q.D.; formal analysis, Q.D. and S.R.; investigation, Q.D., S.R. and L.H.-M.; writing—original draft preparation, Q.D., S.R., L.H.-M. and S.D.; writing—review and editing, S.R., L.H.-M., S.D. and Q.D.; supervision, L.H.-M., S.D. and S.R.; project administration, L.H.-M. and S.D.; funding acquisition, L.H.-M. All authors have read and agreed to the published version of the manuscript.

**Funding:** This study was funded by the French Ministry of Ecology (grant no. 2103207978). Quentin Demarquet received a 2021 Ph.D. grant from MESRI (Ministry of Higher Education, Research and Innovation of France).

**Data Availability Statement:** Not applicable.

**Acknowledgments:** This review is dedicated to the memory of Tom Loveland, thanks to whom the Landsat archive was made freely and globally available, which led to numerous scientific studies and operational tools.

**Conflicts of Interest:** The authors declare no conflict of interest.

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
