# Peer review of "Long-Term Wetland Monitoring Using the Landsat Archive: A Review"

_remotesensing, doi:10.3390/rs15030820_

Round 1

Reviewer 1 Report

The manuscript is good, the authors evaluate the ‘’ Long-term wetland monitoring using the Landsat archive: a review’’. It is an interesting and great contribution to the scientific community; however, the discussion and references of the paper should be improved. Still there are many issues present in the manuscript which should be explained properly. The manuscript needs some minor revisions as given below:

·         The text of this paper in general needs a thorough review, as there are multiple spelling and grammatical errors. Many sentences do not mean any sense. Moreover, there are several sloppy errors that should be fixed.

·         Line 39 to 47, not necessary, please delete it.

·         Manucript is too long and there are some unnecessary contents in the manuscript of this article that can be deleted. It is suggested to modify them carefully and refine the main contents of the article again.

·         ‘’2. Literature search and review’’ should be write as ‘’methodology and analysis’’.

·         In discussion section; Discussion: As per the instruction given by the journal “The findings and their implications should be discussed in the broadest context possible and the limitations of the work highlighted”.

·         Resolution of all figures should be improved.

·         Please add the conclusion in the end of manuscript.

·         Write main results and future recommendation in conclusion.

·         Reference to much, please delete the unnecessary and old references.

·          

Overall, the study conducted is interesting but a minor revision of the entire manuscript is essentially required for publication in this journal. Hence, I recommend reconsideration after a minor revision of the manuscript.  

Author Response

Dear Reviewer,

Thank you for your feedback. Please find attached our response to your comments.

Sincerely,

Quentin Demarquet

Reviewer 2 Report

The paper entitled “Long-term wetland monitoring using the Landsat archive: a review” provided the Landsat data for long-term (for 20 years) wetland monitoring. The result of the review is interesting for the researchers and can be used for future studies. The paper prepared well but needs minor revisions.

1.       Use the keywords that you have in the abstract.

2.       Why do the authors investigate only NDVI? Why did you not consider other vegetation indices such as EVT, SAVI, MSAVI, NDMI, and SR?

3.       Table 4: Why you have not considered the various species in this review?

4.       Figure 5: what are the fit 1, fit 2, …?

5.       It is better to have a section on conclusion.

Author Response

(The authors gave the same response as above.)

Reviewer 3 Report

This paper represents a meta-analysis of literature that used the Landsat archive to identify and assess global wetlands. Across 351 papers that were evaluated, the authors found changes in area (generally a decline), structure (primarily fragmentation and floral differences), and functions. Drivers of change were evaluated and led by human activity, natural hazards, climate change, policies, and/or invasive species. The authors identified issues and opportunities for future work. 

---

STRENGTHS: A well-written paper describing a review of 351 articles relevant to an important ecosystem, wetlands, evaluating what has been reported and the omissions from the literature. 

WEAKNESSES: While an accurate inventory would be a useful product, especially one outlining ecosystem functions, the authors may stretch the application of their results. In addition, I am particularly concerned with the implications of this paper. In the USA, the evaluation of wetland extent has been conducted by the US Fish and Wildlife Service, led by T. Dahl. Because these techniques use products in addition to Landsat, US results appear largely absent from this analysis. If this is true in the US, I question whether it is true in other nations as well. While this paper focuses on Landsat, there is a risk that other important approaches will be ignored. As such, I would appreciate a disclaimer stating something of the sort – in both the paper and the abstract. 

Objections 

The authors note (L30-1) that an “explanation for the decrease and degradation of wetlands could be the absence of a comprehensive and updated wetland inventory”. I do not agree with this premise. An inventory would not curb the development pressures that wetlands experience or any of the drivers that the authors outline as responsible for changes in the wetlands they evaluate. 

The Introduction ends with the authors stating that they conducted a ”systematic review...” of “environmental and remote-sensing issues”. This sets the reader up for higher expectations than are delivered in the paper. Of course, not all issues can be addressed in a single product, so I recommend this be revised to be less broad, even if pointing out that those evaluated are later described in more depth. 

As a wetland ecologist who collaborates with remote sensing experts, I appreciate the note to merge the expertise of both disciplines. In this vein, I find the first paragraph on page 17 to be a shortfall of this paper. Suggesting these particular, individual assessments be used writ large ignores the reason so many approaches have been developed: these systems often have specific functions that are based on their hydrogeomorphic position and the landscape and other factors. As well, many functions cannot be evaluated using the scale of Landsat. Similarly, suggesting that all future studies should use historic MSS imagery (L502-504) ignores the fact that those older images are often misclassified and/or lost at that scale, especially linear features in the landscape. The closing paragraph reiterates what others have said, and minimizes the importance of work conducted on the ground that is not yet suitable to be interpreted at the size of Landsat pixels. 

A final concerns that may be beyond the scope of this paper is that the RAMSAR wetland classification neglects a critical wetland type: tidal freshwater marshes. While a small percentage of the world’s wetlands, these are disappearing from the landscape and worth calling out.  

I request that the Supplemental materials include the references of the 351 papers reviewed. 

Minor copy-editing issues:

L49 insert pixel resolution after near-infra-red for consistency 

L160 “India” is highlighted 

L179 authors are encouraged to at least acknowledge the renaming of the Spartina genus of salt marshes to Sporobolus 

L185-6 Kindly revise what is intended by the upper low marsh – is that the primary area between species of Sporobolus/Spartina? 

L348-9 Perhaps a word is missing? 1749 Landsat data WHAT had the 81-89% accuracy? 

L373 and extra space before the word “used”

Author Response

(The authors gave the same response as above.)

Reviewer 4 Report

The general term “monitoring” is commonly employed in the paper, however the distinction should be made throughout the paper that this study focuses on “long-term” monitoring, as there is an important difference between short-term and long-term monitoring efforts. For example, in line 486 the sentence “wetland functions and ecosystem services have rarely been assessed”, adding in “in long-term monitoring studies” would significantly alter the statement and refine it’s meaning. There are many instances of this throughout the paper.

In general, the bibliography for current studies and mapping efforts could be expanded in order to further emphasize the point and need for long-term monitoring of wetlands at the global scale in the first paragraph. I recommend expanding upon the bibliography for global and tropical wetland extent studies to emphasize that while there have been global mapping initiatives, there is a gap for long-term monitoring efforts. This will highlight the importance of having consistent long-term monitoring in place utilizing available remote sensing data in the first paragraph. For example (but not limited to):

           CIFOR Global Wetlands Map (Tropical) -

https://www2.cifor.org/global-wetlands/

Global Mangrove Watch (Tropical)

https://www.globalmangrovewatch.org/?map=eyJiYXNlbWFwIjoibGlnaHQiLCJ2aWV3cG9ydCI6e319

Global Saltmarsh Extent (2017)

https://bdj.pensoft.net/articles.php?id=11764

PEATMAP: Refining estimates of global peatland distribution based on a meta-analysis

https://www.sciencedirect.com/science/article/abs/pii/S0341816217303004

The map in Figure 2 is very interesting and has the potential to be significantly improved. The composite wetland map by Tootchi et al. show that both tropical and boreal zones hold significant wetland extents through the latitudinal distribution of global wetlands (in Figures 3 and vertically in Figure 13, for example). Given the importance of these two zones for wetlands, and the global wetland distribution, the point can be made to visually highlight the distribution of long-term studies found within this review with the actual distribution of wetlands to determine if the studies are in fact representative of the geospatial wetland distribution, highlight where the broader gaps in knowledge are (for example, are they mostly in the boreal vs tropics?), and make the point for where to prioritize addressing these gaps. While it is mentioned in lines 438-439 that certain countries and regions are underrepresented, these can both be shown graphically improved in Fig. 2 and expanded upon in the text (for example, expanded upon in lines 122-134, and consequent text of the results, discussion, and conclusion). Furthermore, this will strengthen the importance of having openly available data for monitoring efforts in areas in which are lacking, which is of crucial importance in the developing world. While this is mentioned briefly in section 5.3 (lines 505-507), it can be elucidated more as well as better represented in Figure 2.

Lines 524-530. Authors highlight that many images are discarded for processing due to cloud coverage and recommend using all photos available during the study period. However, in tropical areas, which hold globally significant amounts of wetlands, cloud coverage is a significant hindrance that is present much of the year. It is short sited to recommend incorporating all available imagery as this adds significant compute and processing time, for which would not yield any advantage in tropical areas. Furthermore, given developmental and capacity disadvantages of underdeveloped regions, monitoring efforts in the global South are much more resource-limited than in the global North, adding additional difficulties. However, there are globally available composite imagery available from the Landsat archive (8-Day, 32-Day, Annual) that are preprocessed to bypass this challenge presented by cloud coverage (with varying results given the high prevalence of clouds in tropical areas) however the difficulty in accessing these products isn’t their cost (they are freely available on Google Earth Engine – which is in fact not a commercial platform as mentioned in lines 413 and 536) it is in technical difficulties to use cloud compute software. GEE, for example, is an open-source software which is interface through an API using either Python or JavaScript, resulting in a steep technical barrier for entry for data access and processing.

The novelty of this study lies in the quantification of long-term monitoring using the Landsat archive. While this is explained in the text, it is not systematically quantified in a way that is easily understood by the reader – a graphic representation of the results is suggested, which could greatly aid in this communication for the reader. See, for example, Figures 5-9 in Shin et al., 2020 (doi.org/10.3390/f11040368). https://www.mdpi.com/1999-4907/11/4/368

In “Section 4.5. Use of Landsat Sensors” as well as in future recommendations : it would be interesting to quantify the variety of sensors that were used (or not used) in tandem w/ Landsat, given the critical importance of topographic and surface features related to wetlands. In other sections you have mentioned S1/S2 however additional sensors such as DEM or altimetry data (available for 20+ years) can be coupled with spectral sensors, both of which are important for capturing additional topographic or wetland characteristics (such as water height) not captured by spectral data (as shown by Tortini et al., 2020 https://doi.org/10.5194/essd-12-1141-2020 ). Given the topographic importance to many wetland types, there are limitations for solely including spectral data (for example, limitations by cloud coverage in tropical areas, as mentioned above). The addition of non-spectral sensors could be an area for improvement for future studies to improve monitoring efforts, given the importance of the topographic variation of certain wetlands.

Specific recommendations --- 

In all instances of “the GEE” or “the Google Earth Engine”, remove ‘the’ and simply say “GEE” or “Google Earth Engine”

Line 413: You have indicated that GEE is a commercial software. This is false, "Earth Engine is free for noncommercial use" https://earthengine.google.com/faq/.

           Line 536: again a reference to GEE not being open access.

Line 308: change sentence from “and the Landsat archive…” to “combined with the Landsat archive…”

Line 299: Change “such as that available…” to “such as those available…”

Line 151: close the parenthesis.

Line 201: Capitalize the “I” in “indigenous” as a sign of respect, in the same way that English, French, and Spanish (…etc) are capitalized https://www2.gov.bc.ca/gov/content/governments/services-for-government/service-experience-digital-delivery/web-content-development-guides/web-style-guide/writing-guide-for-indigenous-content/capitalization-and-formatting-of-indigenous-terms

Lines 458-459. Add briquetted suggestion to the sentence – “this review revealed that the Landsat archive can also be used to identify drivers of wetland changes [over long-term periods of time].”

Figure 5: not colorblind friendly (e.g. Red/Green)

Tables 2 and 4: Add lines demarcating subsections (such as those present in Table 5)

Table 1: move to supplementary figures.

“Section 4.6 Use of Artificial Intelligence” – While ML techniques are AI, they are generally referred to as Machine Learning. Employing this term will allow your publication to reach a larger audience.

Author Response

(The authors gave the same response as above.)

Reviewer 5 Report

The manuscript “remotesensing-2101321” has reviewed the applications of Landsat archive in wetland monitoring. This topic is interesting and important, as the wetland classification is much difficult than other land use types due to its complex flooding condition. Overall the review was comprehensive and informative. However, I am still wonder several details can be provided before this review can be accepted.

Line 64, before do the literature search, the author should give a definition of “wetland”. The concept of “wetland” is variant due to different purpose. Generally, the “wetland” indicates the transition zone between waterbody and terrestrial. However, the Ramsar definition also include the waterbody such as lake and rivers and marine areas where the depth of water does not exceed six meters. If the lake, river, and marine waters were considered as wetland in this review, the key words in Table 1 should also include the “lake”, “river”, and other water surface.

Line 95-98, I did not find the supplementary material from S1 to S3, please provided.

Line 292-296, there’re many ways to distinguish an area as wetland from other land use types. I prefer if authors can compare the advances and disadvantages of these methods.

Line 492 and 511, the paragraph title of 5.3 and 5.4 are the same with the title of 5.2, please check them.

Author Response

(The authors gave the same response as above.)

Round 2

Reviewer 3 Report

Thank you for responding to my suggestions. I appreciate that you all constrained the applicability of this meta-analysis. The following concerns remain.

(revised) L190:
High and Low marsh are still not clear in this paper. Typically,  high marsh is the area closest to the upland (Sarcocornia spp, Sporobolus pumilus (formerly Spartina patens) & Distichlis spp) whereas the low marsh is the area closest to the creek dominated by Sporobolus alterniflorus (formerly Spartina alterniflora). Both high and ;ow marsh contain Spartina/Spartinobolus species in Virginia (USA).  The reference cited has created an intermediate strip in between the regularly accepted high and low marsh, presumably because in nature the transition is a broad stroke rather than a fine line. The Sun et al. paper is a very small region of Virginia, and this strip is not a widely accepted zone in a salt marsh. Much of the ULM in the  Sun et al. paper was converted to either high or low marsh, with a small amount converting to tidal flat. As written, this evaluation is misleading. It also points out what the authors themselves note is necessary -- and it works both ways. Remote Sensors need to involve ecologists in these types of analyses! The misunderstanding of this particular classification now casts doubts on all of the classifications the team completed.

I was not clear in my original comment regarding RAMSAR wetland classification. RAMSAR does, indeed, recognize tidal freshwater marshes, but the authors do not include this type in Figure 4. RAMSAR recognizes them as class "H" of coastal wetlands (intertidal wetlands). Under which of your groupings do they fall?

paragraph including revised L545:
Many wetland functions CANNOT yet be assessed from space. There is so much intricacy in important features such as soil and microbes that is obscured by the reflectance of the canopy. This fact is completely ignored in this paper. It is not just the large pixels that prevent this; technological limitations still restrict what ecological function can be assigned when the surface is covered by vegetation - even at finer scales. 

-Reviewer #3 (she/her)

Author Response

Dear Reviewer,

Please see in attached our response to your comments.

Sincerely,

Quentin Demarquet

Reviewer 4 Report

The author has made many of the recommended updates to the manuscript and has taken much of the feedback into account. There still stand a few adjustments to be made, however the manuscript has been significantly improved.

Primarily, I don’t think that the case has been made to use every available image in the archive and these statements should be reworded in order to refine the scope as it’s too broad of a conclusion. Yes, while cloud screening and shadow masking is still an important part of filtering and maintaining higher quality time-series, there are still shortcomings with using every single image in a time-series. Some reasons for filtering input data in order to maintain a high quality time series include :

               Reducing the redundancy of information – wetland systems are indeed dynamic, but not that dynamic where they would require monitoring at the temporal resolution of Landsat sensors (2+ times per month). Unless under very specific circumstances, monitoring wetland dynamics, including seasonal water fluctuations or vegetation phenology, does not require multiple images a month. For example, if one is monitoring wetland vegetation it wouldn’t make sense to include periods of time when the vegetation are dormant.

               Maintaining high quality of input data – In addition to the cloud/shadow issues, there are still impurities with pixels and broad sensor issues. For example, the scan line corrector in LT7 broke, rendering many of the images incomplete. It might not necessarily make sense to include these images given the lower quality (or complete lack) of information.

               Processing/compute time and space: even though cloud options do increase the capacity for additional compute space, there are still limits (for example, GEE servers cap out if the user has too high of a compute demand, Amazon’s AWS requires purchase of additional credits for increased computation, etc…). In 2001 alone there are 54 LT8 tiles over the University of Rennes, many of which are unusable due to clouds, and the rest would still require significant compute time and power to process.

Given these circumstances, the case isn’t made to use all images in the archive for long term monitoring and it therefore still makes sense to filter your timeseries down. However given the advancements in cloud computing and modeling I could see the argument being made to increase the amount of available input data to capture intra-annual variability, if the data had been excluded due to computation limitations.

These are specifically mentioned in the following sentences: “…processing of all available images (lines 314-315)…Future research should focus on modeling the temporal profile using all of the images available in the Landsat archive. Cloud removal methods are now available that allow the full Landsat time series to be used over the entire globe, which can improve long-term monitoring, including in tropical areas (lines 593-594)…focus on modeling the temporal profile using all of the images available in the Landsat archive (lines 641-642)”

Additional changes

               Figures 5 & 6 : reorder stacked bars from high to low (e.g., “Functions” in Fig. 5 should be : 72%, 15%, 11%, then 2% last)

               Line 384: reword to “…Landsat data with an overall classification accuracy of…”

               Line 455: “Cloud-computing studies were published more often in remote sensing journals”. Specify how much more.

               Line 456: Remove the ‘s’ in “for examples”

               Line 463: delete ‘the’ in “including the NDVI”

               Table 2: I am unclear on the meaning of this table, specifically with what “class” means, in addition to the explanation ““Class/Mod”: studies with the attribute and belonging to the class; "Mod/Class”: studies belonging to the class and with the attribute”

               Line 542 : add in bracketed suggestion to the sentence “rarely been assessed [in long-term monitoring]”

               Line 603 : reword to “…gaps between countries in the Global North and South…”. I think the overall language you’re looking for here is more along the lines of data democratization brought through the scalability that cloud compute offers.

               Lines 603-604 : delete “although specific computing skills and learning Python and JavaScript programming languages are needed”

               Line 614, Fig. 8 subtext : “The fits in the colored lines…”

               Line 607 – Microsoft Planetary Computer is an additional cloud based open-source platform

Future preference – my pronouns are she/her (when in doubt it’s safe to default to the gender-neutral pronouns they/their)

Author Response

(The authors gave the same response as above.)
